# ON THE IMPORTANCE AND APPLICABILITY OF PRE-TRAINING FOR FEDERATED LEARNING

**Hong-You Chen, Cheng-Hao Tu, Ziwei Li, Han-Wei Shen, Wei-Lun Chao**
Department of Computer Science and Engineering, The Ohio State University

## ABSTRACT

Pre-training is prevalent in nowadays deep learning to improve the learned model's performance. However, in the literature on federated learning (FL), neural networks are mostly initialized with random weights. These attract our interest in conducting a systematic study to explore pre-training for FL. Across multiple visual recognition benchmarks, we found that pre-training can not only improve FL, but also close its accuracy gap to the counterpart centralized learning, especially in the challenging cases of non-IID clients' data. To make our findings applicable to situations where pre-trained models are not directly available, we explore pre-training with synthetic data or even with clients' data in a decentralized manner, and found that they can already improve FL notably. Interestingly, many of the techniques we explore are complementary to each other to further boost the performance, and we view this as a critical result toward scaling up deep FL for real-world applications. We conclude our paper with an attempt to understand the effect of pre-training on FL. We found that pre-training enables the learned global models under different clients' data conditions to converge to the same loss basin, and makes global aggregation in FL more stable. Nevertheless, pre-training seems to not alleviate local model drifting, a fundamental problem in FL under non-IID data.

## 1 INTRODUCTION

The increasing attention to data privacy and protection has attracted significant research interests in federated learning (FL) (Li et al., 2020a; Kairouz et al., 2019). In FL, data are kept separate by individual clients. The goal is thus to learn a "global" model in a decentralized way. Specifically, one would hope to obtain a model whose accuracy is as good as if it were trained using centralized data.

FEDAVG (McMahan et al., 2017) is arguably the most widely used FL algorithm, which assumes that every client is connected to a server. FEDAVG trains the global model in an *iterative* manner, between parallel *local model training* at the clients and *global model aggregation* at the server. FEDAVG is easy to implement and enjoys theoretical guarantees of convergence (Zhou & Cong, 2017; Stich, 2019; Haddadpour & Mahdavi, 2019; Li et al., 2020c; Zhao et al., 2018). Its performance, however, can degrade drastically when clients' data are not IID — clients' data are often collected individually and doomed to be non-IID. That is, the accuracy of the federally learned global model can be much lower than its counterpart trained with centralized data. To alleviate this issue, existing literature has explored better approaches for local training (Li et al., 2020b; Karimireddy et al., 2020b; Acar et al., 2021) and global aggregation (Wang et al., 2020a; Hsu et al., 2019; Chen & Chao, 2021).

In this paper, we explore a different and rarely studied dimension in FL — *model initialization*. In the literature on FL, neural networks are mostly initialized with random weights. Yet in centralized learning, model initialization using weights pre-trained on large-scale datasets (Hendrycks et al., 2019; Devlin et al., 2018) has become prevalent, as it has been shown to improve accuracy, generalizability, robustness, etc. *We are thus interested in 1) whether model pre-training is applicable in the context of FL and 2) whether it can likewise improve* FEDAVG, *especially in alleviating the non-IID issue.*

We conduct the very first *systematic study* in these aspects, using visual recognition as the running example. We consider multiple application scenarios, with the aim to make our study comprehensive.

First, assuming pre-trained weights (e.g., on ImageNet (Deng et al., 2009)) are available, we systematically compare FEDAVG initialized with random and pre-trained weights, under different FL settings

and across multiple visual recognition tasks. These include four image classification datasets, CIFAR-10/100 (Krizhevsky et al., 2009), Tiny-ImageNet (Le & Yang, 2015), and iNaturalist (Van Horn et al., 2018), and one semantic segmentation dataset, Cityscapes (Cordts et al., 2016). We have the following major observations. We found that pre-training consistently improves FEDAVG; the relative gain is more pronounced in more challenging FL settings (e.g., severer non-IID conditions across clients). Moreover, pre-training largely closes the accuracy gap between FEDAVG and centralized learning (Figure 1), suggesting that pre-training brings additional benefits to FEDAVG than to centralized learning. We further consider more advanced FL methods (e.g., (Li et al., 2020b; Acar et al., 2021; Li et al., 2021b)). We found that pre-training improves their accuracy but diminishes their gain against FEDAVG, suggesting that FEDAVG is still a strong FL approach if pre-trained weights are available.

Second, assuming pre-trained models are not available and there are no real data at the server for pre-training, we explore the use of *synthetic data*. We investigate several simple yet effective synthetic image generators (Baradad et al., 2021), including fractals which are shown to capture geometric patterns found in nature (Mandelbrot & Mandelbrot, 1982). We propose a new pre-training scheme called *Fractal Pair Similarity (FPS)* inspired by the inner workings of fractals, which can consistently improve FEDAVG for the downstream FL tasks. This suggests the wide applicability of pre-training to FL, even without real data for pre-training.

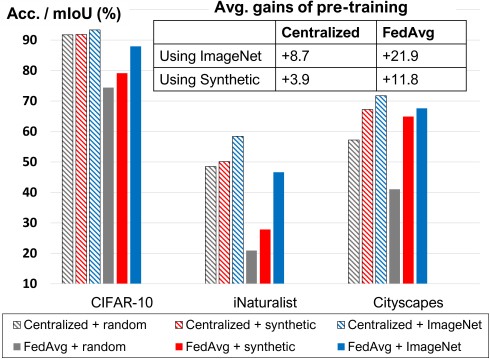

Third, we explore the possibility to directly pre-train with clients' data. Specifically, we investigate the two-stage training procedure — self-supervised pre-training, followed by supervised learning — in a federated setting. Such a procedure has been shown to outperform pure supervised learning in centralized learning (Chen et al., 2020c), but has not been explored in FL. Using the state-of-the-art federated self-supervised approach (Lubana et al., 2022), we not only demonstrate its effectiveness in FL, but also show its compatibility with available pre-trained weights to further boost the performance.

Figure 1: **Pre-training improves FEDAVG more than it improves centralized learning.** We consider three initialization weights: random, pre-trained on ImageNet, and pre-trained on synthetic images. Pre-training helps both FEDAVG and centralized learning, but has a larger impact on FEDAVG. Even without real data, our proposed pre-training with synthetic data is sufficient to improve FEDAVG notably.

Intrigued by the improvement brought by pre-training, we make an attempt to understand its underlying effect on FL. We first analyze the training dynamics of FEDAVG. We found that pre-training seems to not alleviate local model drifting (Li et al., 2020b; Karimireddy et al., 2020b), a well-known issue under non-IID data. Nevertheless, it makes global aggregation more stable. Concretely, FEDAVG combines the local models' weights simply by coefficients proportional to local data sizes. Due to model drifting in local training, these coefficients can be far from optimal (Chen & Chao, 2021). Interestingly, with pre-training, FEDAVG is less sensitive to the coefficients, resulting in a stronger global model in terms of accuracy. Through visualizations of the loss landscapes (Li et al., 2018; Hao et al., 2019), we further found that pre-training enables the learned global models under different data conditions (i.e., IID or various non-IID degrees) to converge to the same loss basin. Such a phenomenon can hardly be achieved without pre-training, even if we initialize FEDAVG with the same random weights. This offers another explanation of why pre-training improves FL.

**Contributions and scopes.** We conduct the very first systematic study on pre-training for FL, including a novel synthetic data generator. We believe that such a study is timely and significant to the FL community. We focus on visual recognition using five image datasets. We go beyond them by further studying semantic segmentation problems on the Cityscape dataset, not merely classification problems. Our extended analyses reveal new insights into FL, opening up future research directions.

## 2 RELATED WORK

**Federated learning (FL).** FEDAVG (McMahan et al., 2017) is the fundamental FL algorithm. Many works were proposed to improve it, especially to alleviate its accuracy drop under non-IID data. In *global aggregation*, Wang et al. (2020a); Yurochkin et al. (2019) matched local model weights before

averaging. Lin et al. (2020); He et al. (2020a); Zhou et al. (2020); Chen & Chao (2021) replaced weight average by model ensemble and distillation. Hsu et al. (2019); Reddi et al. (2021) applied server momentum and adaptive optimization. In *local training*, to reduce local model drifting — a problem commonly believed to cause the accuracy drop —Zhao et al. (2018) mixed client and server data; FEDPROX (Li et al., 2020b), FEDDANE (Li et al., 2019), and FEDDYN (Acar et al., 2021) employed regularization toward the global model; SCAFFOLD (Karimireddy et al., 2020a) and MIME (Karimireddy et al., 2020b) used control varieties or server statistics to correct local gradients; Wang et al. (2020b); Yao et al. (2019) modified local update rules.

We investigate a rarely studied aspect to improve FEDAVG, *initialization*. To our knowledge, very few works in FL have studied this (Lin et al., 2022; Stremmel & Singh, 2021; Weller et al., 2022); none are as systematic and comprehensive as ours[1]. (Qu et al., 2021; Hsu et al., 2020; Cheng et al., 2021) used pre-trained models in their experiments but did not or only briefly analyzed their impacts.

**Pre-training.** Pre-training has been widely applied in computer vision, natural language processing, and many other application domains to speed up convergence and boost accuracy for downstream tasks (Kolesnikov et al., 2020; Goyal et al., 2021; Radford et al., 2021; Sun et al., 2017; Devlin et al., 2018; Yang et al., 2019; Brown et al., 2020). Many works have attempted to analyze its impacts (Hendrycks et al., 2020; Erhan et al., 2010; He et al., 2019; Djolonga et al., 2021; He et al., 2019; Kornblith et al., 2019). For example, Hendrycks et al. (2019) and Chen et al. (2020a) found that pre-training improves robustness against adversarial examples; Neyshabur et al. (2020) studied the loss landscape when fine-tuning on target tasks; Mehta et al. (2021) empirically showed that pre-training reduces forgetting in continual learning. Despite the ample research on pre-training, its impacts on FL remain largely unexplored. We aim to fill in this missing piece.

## 3 BACKGROUND: FEDERATED LEARNING

In **federated learning (FL)**, the training data are collected by $M$ clients. Each client $m \in [M]$ has a training set $\mathcal{D}_m = \{(\boldsymbol{x}_i, y_i)\}_{i=1}^{|\mathcal{D}_m|}$, where $\boldsymbol{x}$ is the input (e.g., images) and $y$ is the true label. The goal is to solve the following optimization problem

$$\min_{\boldsymbol{\theta}} \ \mathcal{L}(\boldsymbol{\theta}) = \sum_{m=1}^{M} \frac{|\mathcal{D}_m|}{|\mathcal{D}|} \mathcal{L}_m(\boldsymbol{\theta}), \quad \text{where} \quad \mathcal{L}_m(\boldsymbol{\theta}) = \frac{1}{|\mathcal{D}_m|} \sum_{i=1}^{|\mathcal{D}_m|} \ell(\boldsymbol{x}_i, y_i; \boldsymbol{\theta}). \tag{1}$$

Here, $\boldsymbol{\theta}$ is the model parameter; $\mathcal{D} = \cup_m \mathcal{D}_m$ is the aggregated training set from all clients; $\mathcal{L}$ is the empirical risk on $\mathcal{D}$; $\mathcal{L}_m$ is the empirical risk of client $m$; $\ell$ is the loss function on a data instance.

**Federated averaging (FEDAVG).** As clients' data are separated, Equation 1 cannot be solved directly; otherwise, it is **centralized learning**. A standard way to relax it is FEDAVG (McMahan et al., 2017), which iterates between two steps — parallel *local training* at the clients and *global aggregation* at the server — for multiple rounds of communication

$$\textbf{Local:} \ \tilde{\boldsymbol{\theta}}_m^{(t)} = \arg\min_{\boldsymbol{\theta}} \mathcal{L}_m(\boldsymbol{\theta}), \text{ initialized by } \bar{\boldsymbol{\theta}}^{(t-1)}; \qquad \textbf{Global:} \ \bar{\boldsymbol{\theta}}^{(t)} \leftarrow \sum_{m=1}^{M} \frac{|\mathcal{D}_m|}{|\mathcal{D}|} \tilde{\boldsymbol{\theta}}_m^{(t)}. \tag{2}$$

The superscript $t$ indicates the models after round $t$; $\bar{\boldsymbol{\theta}}^{(0)}$ denotes the initial model. That is, local training aims to minimize each client's empirical risk, often by several epochs of stochastic gradient descent (SGD). Global aggregation takes an element-wise average over local model parameters.

**Problem.** When clients' data are non-IID, $\tilde{\boldsymbol{\theta}}_m^{(t)}$ would drift away from each other and from $\bar{\boldsymbol{\theta}}^{(t-1)}$, making $\bar{\boldsymbol{\theta}}^{(t)}$ deviate from the solution of Equation 1 and resulting in a drastic accuracy drop.

## 4 PRE-TRAINING IS APPLICABLE TO AND IMPORTANT FOR FL

In most of the FL literature that learns neural networks, $\bar{\boldsymbol{\theta}}^{(0)}$ is initialized with random weights. We thus aim to provide a detailed and systematic study on pre-training for FL. Specifically, we are interested in whether pre-training helps FEDAVG alleviate the accuracy drop in non-IID conditions.

---

[1]A concurrent work by Nguyen et al. (2022) presents an empirical study as well. They focus more on how pre-training affects federated optimization algorithms. We study both the impact and applicability of pre-training for FL, provide further analyses and insights to understand them, and evaluate on large-scale datasets.

## 4.1 PRE-TRAINING SCENARIOS IN THE CONTEXT OF FL

However, to begin with, we must consider if pre-training is feasible for FL. Namely, can we obtain pre-trained weights in FL applications? We consider two scenarios. **First**, the server has pre-trained weights or data for pre-training; **second**, the server has none of them. For instance, in areas like computer vision, many pre-trained models and large-scale datasets are publicly accessible. However, for data-scarce or -costly domains, or privacy-critical domains like medicine, these resources are often not publicly accessible. *For the second scenario, we explore the use of synthetic data for pre-training.*

**Running example.** Throughout the paper, we use visual recognition as the running example to study both scenarios, even though in reality we usually have pre-trained models for it. The rationales are two-folded. First, it makes our study coherent, i.e., reporting the results on the same tasks. Second, it allows us to assess the gap between collecting real or creating synthetic data for pre-training.

**Scenario one.** We mainly use weights pre-trained on ImageNet (1K) (Russakovsky et al., 2015; Deng et al., 2009). For downstream FL tasks derived from ImageNet (e.g., Tiny-ImageNet (Le & Yang, 2015)), we use weights pre-trained on Places365 (Zhou et al., 2017) to prevent data leakage. For both cases, pre-training is done by standard supervised training.

**Scenario two.** We use synthetic data for pre-training. We consider image generators like random generative models (Baradad et al., 2021) and fractals (Kataoka et al., 2020) that are not tied to specific tasks. These generators, while producing non-realistic images, have been shown effective for pre-training. We briefly introduce fractals, as they achieve the best performance in our study.

A fractal image can be generated via an affine Iterative Function System (IFS) (Barnsley, 2014). An IFS generates a fractal by "drawing" points iteratively on a canvas. The point transition is governed by a small set of $2 \times 2$ affine transformations (each with a probability), denoted by $\mathcal{S}$. Concretely, given the current point $v_n \in \mathbb{R}^2$, the IFS randomly samples one affine transformation with replacement from the set $\mathcal{S}$, and uses it to transform $v_n$ into the next point $v_{n+1}$. This process continues until a sufficient number of iterations is reached. The collection of points can then be used to draw a fractal image: by rendering each point as a binary or continuous value on a black canvas. Since $\mathcal{S}$ controls the generation process and hence the geometry of the fractal, it can essentially be seen as the fractal's ID, code, or class label; different codes would generate different fractals. Due to the randomness in the IFS, the same code $\mathcal{S}$ can create different but geometrically-consistent fractals.

Using these properties of fractals, Kataoka et al. (2020) proposed to sample $J$ different codes, create for each code a number of images, and pre-train a model in a supervised way. Anderson & Farrell (2022) proposed to create more complex images by painting $I$ fractals on one canvas. The resulting image thus has $I$ out of $J$ classes, and can be used to pre-train a model via a multi-label loss.

In our study, we however found none of them effective for FL, perhaps due to their limitation on $J$. We thus propose a novel way for fractal pre-training, inspired by one critical insight — *we can in theory sample infinitely many IFS codes and create fractals with highly diverse geometry*. In the extreme, every image can be synthesized with an entirely different code.

We propose to pre-train with such fractal images using contrastive or similarity learning (Chen et al., 2020d; Chen & He, 2021). The idea is to treat every image as from a different class and learn to repulse different images away (i.e., negative pairs) or draw an image and its augmented version closer (i.e., positive pairs). We propose a unique way to create positive pairs, which is to use IFS to create a pair of images based on the same codes. We argue that this leads to stronger and physically-meaningful argumentation: not only do different fractals from the same codes capture intra-class variation, but we also can create a pair of images with different placements of the same fractals (more like object-level augmentation). We name our approach **fractal pair similarity (FPS)** (see Figure 2).

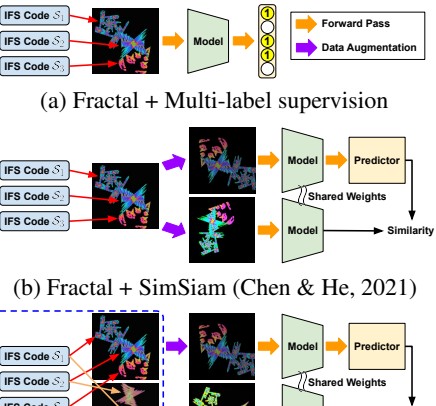

(a) Fractal + Multi-label supervision

(b) Fractal + SimSiam (Chen & He, 2021)

(c) Fractal Pair Similarity (ours) + SimSiam

Figure 2: **Pre-training with fractals.** (a) multi-label training; (b) similarity learning; (c) similarity learning with our fractal pair similarity (FPS).

## 4.2 EXPERIMENTAL SETUP AND IMPLEMENTATION DETAILS

**Data.** We conduct the study using five visual recognition datasets: CIFAR-10, CIFAR-100 (Krizhevsky et al., 2009), Tiny-ImageNet (Le & Yang, 2015), iNaturalist (Van Horn et al., 2018), and Cityscapes (Cordts et al., 2015). The first four are for image classification; the last is for semantic

Table 1: Summary of datasets and setups.

| Dataset | #Class | #Training | #Test | #Clients $M$ | Resolution |
|---------|--------|-----------|-------|--------------|------------|
| CIFAR-10/100 | 10/100 | $50K$ | $10K$ | $10 - 100$ | $32^2$ |
| Tiny-ImageNet | 200 | $100K$ | $10K$ | $10 - 1000$ | $64^2$ |
| iNaturalist-GEO | 1203 | $120K$ | $36K$ | $39/136$ | $224^2$ |
| Cityscapes | 19 | $3K$ | $0.5K$ | 18 | $768^2$ |

segmentation. Table 1 summarizes their statistics. (*We also include NLP tasks such as sentiment analysis (Caldas et al., 2018) and language modeling (McMahan et al., 2017) in the Suppl.*)

**Non-IID splits.** To simulate non-IID conditions across clients, we follow (Hsu et al., 2019) to partition the training set of CIFAR-10, CIFAR-100, and Tiny-ImageNet into $M$ clients. To split the data of class $c$, we draw an $M$-dimensional vector $\boldsymbol{q}_c$ from Dirichlet($\alpha$), and assign data of class $c$ to client $m$ proportionally to $\boldsymbol{q}_c[m]$. The resulting clients have different numbers of total images and class distributions. The small the $\alpha$ is, the more severer the non-IID condition is.

For iNaturalist, a dataset for species recognition, we use the data proposed by (Hsu et al., 2020), which are split by *Geo locations*. There are two versions, GEO-10K (39 clients) and GEO-3K (136 clients). For Cityscapes, a dataset of street scenes, we use the official training/validation sets that contain 18/3 cities in Germany. We split the training data by *cities* to simulate a realistic scenario.

**Metrics.** We report the averaged **accuracy** (%) for classification and the **mIoU** (%) for segmentation on the global test set. For brevity, we denote the test accuracy or mIoU of the model $\boldsymbol{\theta}$ by $\mathcal{A}_{\text{test}}(\boldsymbol{\theta})$.

**Models.** We use ResNet20 (He et al., 2016a) for CIFAR-10/100, which is suitable for a $32 \times 32$ resolution. We use ResNet18 for Tiny-ImageNet and iNaturalist. For semantic segmentation, we use DeepLabV3+ (Chen et al., 2018) with a MobileNet-V2 (Sandler et al., 2018) backbone.

**Initialization.** We compare random and pre-trained weights. For scenario one (subsection 4.1), we obtain pre-trained weights for ResNet18 and MobileNet-V2 from PyTorch's and Places365's official sites. For ResNet20, we pre-train the weights using down-sampled, $32 \times 32$ ImageNet images.

For scenario two using synthetic images, we mainly consider fractals and apply our FPS approach, but will investigate other approaches in subsection 4.5. We apply the scheme proposed by Anderson & Farrell (2022) to sample IFS codes (each with $2 \sim 4$ transformations). For efficiency, we pre-sample a total of 100K IFS codes, and uniformly sample $I$ codes from them to generate each fractal image. We then follow Anderson & Farrell (2022) to color, resize, and flip the fractals. We then apply a similarity learning approach SimSiam (Chen & He, 2021) for pre-training, due to its efficiency and effectiveness. We pre-train the weights for 100 epochs; each epoch has 1M image pairs (or equivalently 2M images). See Figure 2 for an illustration and see the Suppl. for details.

More specifically, for ResNet20, we set $I = 2$ and run the IFS for 1K iterations to render one $32 \times 32$ image. For ResNet18 and DeepLabV3+, we sample $I$ uniformly from $\{2, 3, 4, 5\}$ for each image to increase diversity, and run the IFS for 100K iterations to render one $224 \times 224$ image.

**Federated learning.** We perform FEDAVG for **100 iterative rounds**. Each round of local training takes 5 epochs[2]. We use the SGD optimizer with weight decay $1e-4$ and a $0.9$ momentum, except that on DeepLabV3+ we use an Adam (Kingma & Ba, 2015) optimizer. We apply the standard image pre-processing and data augmentation (He et al., 2016a). We reserve $2\%$ data of the training set as the validation set for hyperparameter tuning (e.g., for the learning rate[3]). We follow the literature (He et al., 2016b) to decay the learning rate by $0.1$ every 30 rounds. We leave more details and the selected hyperparameters in the Suppl.

## 4.3 EMPIRICAL STUDY ON CIFAR-10 AND TINY-IMAGENET

We first focus on CIFAR-10 and Tiny-ImageNet, on which we create artificial splits of clients' data to simulate various non-IID conditions. Besides applying FEDAVG to train a model $\boldsymbol{\theta}_{\text{FL}}$ in an FL way, we also aggregate clients' data and train the corresponding model $\boldsymbol{\theta}_{\text{CL}}$ in a centralized learning way. This enables us to assess their accuracy gap $\Delta_{\text{CL-FL}} = \mathcal{A}_{\text{test}}(\boldsymbol{\theta}_{\text{CL}}) - \mathcal{A}_{\text{test}}(\boldsymbol{\theta}_{\text{FL}})$.

---

[2]We found this number quite stable for all experiments and freeze it throughout the paper.

[3]We use the same learning rate within each dataset, which is selected by FEDAVG without pre-training.

Table 2: Test accuracy on CIFAR-10 ($\alpha = 0.3$).

| Initialization | Federated | Centralized | $\Delta_{\text{CL-FL}}$ |
|---|---|---|---|
| Random | 82.7 | 91.7 | 9.0 |
| FPS | 84.9 (+2.2) | 92.0 (+0.3) | 7.1 |
| ImageNet | 90.8 (+8.1) | 93.3 (+1.6) | 2.5 |

Table 3: Test accuracy on Tiny-ImageNet ($\alpha = 0.3$).

| Initialization | Federated | Centralized | $\Delta_{\text{CL-FL}}$ |
|---|---|---|---|
| Random | 42.4 | 47.5 | 5.1 |
| FPS | 45.7 (+3.3) | 49.1 (+1.6) | 3.4 |
| Places365 | 50.3 (+7.5) | 52.9 (+2.8) | 2.6 |

(a) #Clients    (b) Dir($\alpha$)-non-IID    (c) Participation (%)    (d) #Local epochs/round

Figure 3: Comparison of model initialization across different federated settings. The default setting is 10 clients, $\alpha = 0.3$, 100% participation, and 5 local epochs, and we change one variable at a time.

**Pre-training improves FEDAVG and closes its gap to centralized learning.** We first set $\alpha = 0.3$ (a mild non-IID degree) and $M = 10$ (i.e., 10 clients), and study the case that all clients participate in every round. Table 2 and Table 3 summarize the results; each row is for different initialization. As shown in the column "Federated," pre-training consistently improves the accuracy of FEDAVG; the gain is highlighted by the magenta digits. This is not only by pre-training using large-scale real datasets like ImageNet and Places365, but also by pre-training using synthetic images.

We further investigate whether pre-training helps bridge the gap between FEDAVG and centralized learning. As shown in the column "Centralized," pre-training also improves centralized learning. However, by looking at the column "$\Delta_{\text{CL-FL}}$," which corresponds to the difference between "Federated" and "Centralized," we found that pre-training closes their gap. The gap on CIFAR-10 was 9.0 without pre-training, and reduces to 2.5 with weights pre-trained on ImageNet. In other words, pre-training seems to bring more benefits to FL than to centralized learning.

**Pre-training brings larger gains to more challenging settings.** We now consider different federated settings. This includes different numbers of clients $M$, different non-IID degrees $\alpha$, and different percentages of participating clients per round. We also consider different numbers of local training epochs per round, but keep the total local epochs in FEDAVG as 500. For each configuration, we conduct three times of experiments and report the mean and standard deviation.

Figure 3 summarizes the results, in which we change one variable at a time upon the default setting: $M = 10$, $\alpha = 0.3$, 100% participation, and 5 local epochs. Across all the settings, we see robust gains by pre-training, either using real or synthetic images. *Importantly, when the setting becomes more challenging, e.g., smaller $\alpha$ for larger non-IID degrees or larger $M$ for fewer data per client, the gain gets larger.* This shows the value of pre-training in addressing the challenge in FL.

**Pre-training is compatible with other FL algorithms.** We now consider advanced FL methods like FEDPROX (Li et al., 2020b), FEDDYN (Acar et al., 2021), and MOON (Li et al., 2021b). They were proposed to improve FEDAVG, and we want to investigate if pre-training could still improve them. Figure 4 shows the results, using the default federated setting. We found that both pre-training with real (right bars) and

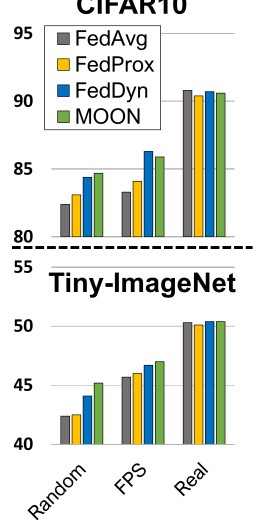

Figure 4: Pre-training for different FL methods.

synthetic (middle bars) images can consistently boost their accuracy. Interestingly, when pre-training with real data is considered, all these FL methods, including FEDAVG, perform quite similarly. This suggests that FEDAVG is still a strong FL approach if pre-trained models are available.

**Pre-training makes network sizes scale better.** One challenge in FL is the difficulty to train deeper networks (Chen & Chao, 2021). We investigate if pre-training mitigates this issue. On CIFAR-10 ($\alpha = 0.3$), we study different network depths and widths based on ResNet20. As shown in Figure 5, with random initialization, deeper models have quite limited gains; wider models improve more. With pre-training (either FPS or real), both going deeper and wider have notable gains, suggesting that pre-training makes training larger models easier in FL.

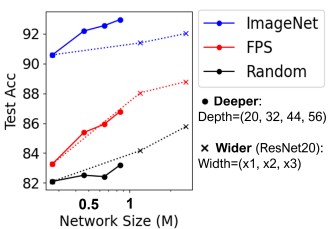

Figure 5: Deeper/wider ResNet.

## 4.4 EMPIRICAL STUDY ON LARGE-SCALE AND REAL DATA SPLITS

We now study pre-training for FL on large-scale datasets, iNaturalist 2017 (Van Horn & Perona, 2017) and Cityscapes (Cordts et al., 2016). Both datasets provide geo-location information, and can be split in a realistic way to simulate location-specific clients. (Please see subsection 4.2 for details.)

Table 4 summarizes the results on iNaturalist 2017 (Van Horn & Perona, 2017), using the *GEO-10K* (39 clients and $50\%$ clients per round) and *GEO-3K* (136 clients and $20\%$ clients per round) splits proposed by (Hsu et al., 2020).

Table 4: Test accuracy on iNaturalist-GEO (Hsu et al., 2020)

| Init. | GEO-10K | GEO-3K | Centralized | Avg. $\Delta_{\text{CL-FL}}$ |
|---|---|---|---|---|
| Random | 20.9 | 12.2 | 48.5 | 32.0 |
| FPS | 27.8 (+6.9) | 17.7 (+5.5) | 50.2 (+1.7) | 27.5 |
| ImageNet | 46.6 (+25.7) | 45.6 (+33.4) | 58.4 (+9.9) | 12.3 |

Table 5 summarizes the semantic segmentation results (mIoU) on Cityscapes (Cordts et al., 2016). The training data are split into 18 clients by cities. We consider full client participation at every round.

Table 5: mIoU (%) of segmentation on Cityscapes.

| Init. / | Federated | Centralized | $\Delta_{\text{CL-FL}}$ |
|---|---|---|---|
| Random | 41.0 | 57.2 | 16.2 |
| FPS | 64.9 (+23.9) | 67.2 (+10.0) | 2.3 |
| ImageNet | 67.6 (+26.6) | 71.1 (+13.9) | 3.5 |

In both tables, we see clear gaps between centralized and federated learning (i.e., $\Delta_{\text{CL-FL}}$) on realistic non-IID splits, and pre-training with either synthetic (i.e., FPS) or real data notably reduces the gaps. Specifically, compared to random initialization, FPS brings encouraging gains to FEDAVG ($> 5\%$ on iNaturalist; $> 20\%$ on Cityscapes). The gain is larger than that of centralized learning.

## 4.5 ON PRE-TRAINING WITH SYNTHETIC DATA

In previous subsections, we show that even without available pre-trained models or real data for pre-training, we can resort to pre-training with synthetic data and still achieve a notable gain for FL. Here, we provide more analyses and discussions.

Table 6: Comparison on synthetic pre-training. Means of 3 runs are reported.

| Init. | C10 | C100 | Tiny |
|---|---|---|---|
| Random | 74.4 | 51.4 | 42.4 |
| Fractal + Multi-label | 73.0 | 51.0 | 40.9 |
| StyleGAN + SimSiam | 79.2 | 53.0 | 44.6 |
| Fractal + SimSiam | 77.4 | 51.7 | 44.2 |
| **FPS (ours)** + SimSiam | **80.5** | **54.7** | **45.7** |

**Comparison of synthetic pre-training methods.** For fractals, we compare the three methods in Figure 2. All of them use the same setup in subsection 4.2, except for multi-label learning we choose $J = $1K IFS codes as it gives the best validation accuracy. We also consider synthetic images generated by (the best version of) random StyleGAN (Baradad et al., 2021; Karras et al., 2019). We use these methods to initialize FEDAVG on CIFAR-10/100 ($\alpha = 0.1$, large non-IID) and Tiny-ImageNet ($\alpha = 0.3$), with $M = 10$ clients and full participation. Results are in Table 6. Our approach FPS outperforms all the baselines. By taking a deeper look, we found that pre-training with multi-label supervision does not outperform random initialization. We attribute this to both the limited diversity of fractals and the learning mechanism: we tried to increase $J$ but cannot improve due to poor convergence. Self-supervised learning, on the contrary, can better learn from diverse fractals (i.e., each fractal a class). By taking the inner working of fractals into account to create geometrically-meaningful positive pairs, our FPS unleashes the power of synthetic pre-training.

**Privacy-critical domain.** We conduct a study on chest X-ray diagnosis (Kermany et al., 2018). Such a medical image domain is privacy-critical, and has a large discrepancy from ImageNet images. Under the same setup as in subsection 4.3 ($\alpha = 0.3$, $M = 10$, full participation), FEDAVG initialized with random/FPS/ImageNet achieves $68.9/74.8/69.1\%$ test accuracy. Interestingly, synthetic pre-training outperforms ImageNet pre-training by $5.9\%$. This showcases when synthetic data can be practically useful in FL: when the problem domain is far from where accessible pre-trained models are trained.

Figure 6: **Training dynamics of FEDAVG.** Please see the text in section 6 for details.

## 5 FEDERATED SELF-PRE-TRAINING CAN IMPROVE FL

Seeing the benefit of pre-training, we investigate another way to obtain a good representation to initialize FL tasks, which is to perform *self-supervised learning* directly on clients' decentralized data. Self-supervised learning (Liu et al., 2021) has been shown powerful to obtain good representations, which sometimes even outperform those obtained by supervised learning (Ericsson et al., 2021). Chen et al. (2020c) further showed that a two-stage self-pre-training (SP) procedure — self-supervised pre-training, followed by supervised learning, on the same data — can outperform pure supervised learning.

Table 7: Self-pre-training (**SP**) for FL.

| Init. | |SP | C10 | Tiny |
|---|---|---|---|
| Random | ✗ | 74.4 | 42.4 |
| | ✓ | 79.8 | 43.9 |
| FPS | ✗ | 79.9 | 45.7 |
| | ✓ | 82.2 | 46.0 |
| Real | ✗ | 87.9 | 50.3 |
| | ✓ | 88.0 | 50.6 |

We explore such an idea in FL. Starting from the random weights (or even pre-trained weights), we first ignore the labels of clients' data and apply a state-of-the-art federated self-supervised learning algorithm (Lubana et al., 2022) (for 50 rounds), followed by standard FEDAVG. We use the same setup as in subsection 4.5 for CIFAR-10 ($\alpha = 0.1$) and Tiny-ImageNet ($\alpha = 0.3$), and summarize the results in Table 7. The SP procedure can significantly improve FEDAVG initialized with random weights, without external data. (We confirmed the gain is not merely from that in total we perform more rounds of FL. We extended FEDAVG without SP for 50 rounds but did not see an improvement.) Interestingly, the SP procedure can also improve upon pre-trained weights. These results further demonstrate the importance of initialization for FEDAVG. (See the Suppl. for more discussions.)

## 6 AN ATTEMPT TO UNDERSTAND THE EFFECTS, AND CONCLUSION

We take a deeper look at pre-training (using real data) for FL and provide further analyses.

**Preparation and notation.** We first identify factors that may affect FEDAVG's accuracy along its training process. Let us denote by $\mathcal{D}_{\text{test}} = \{(\boldsymbol{x}_i, y_i)\}_{i=1}^{N_{\text{test}}}$ the global test data, by $\mathcal{L}_{\text{test}}(\boldsymbol{\theta})$ the test loss, and by $\mathcal{A}_{\text{test}}(\boldsymbol{\theta})$ the test accuracy. Following Equation 2, we decompose $\mathcal{A}_{\text{test}}(\bar{\boldsymbol{\theta}}^{(t)})$ after round $t$ by

$$\mathcal{A}_{\text{test}}(\bar{\boldsymbol{\theta}}^{(t-1)}) + \underbrace{\sum_{m=1}^{M} \frac{|\mathcal{D}_m|}{|\mathcal{D}|} \mathcal{A}_{\text{test}}(\tilde{\boldsymbol{\theta}}_m^{(t)}) - \mathcal{A}_{\text{test}}(\bar{\boldsymbol{\theta}}^{(t-1)})}_{\Delta_L^{(t)}} + \underbrace{\mathcal{A}_{\text{test}}(\bar{\boldsymbol{\theta}}^{(t)}) - \sum_{m=1}^{M} \frac{|\mathcal{D}_m|}{|\mathcal{D}|} \mathcal{A}_{\text{test}}(\tilde{\boldsymbol{\theta}}_m^{(t)})}_{\Delta_G^{(t)}}. \quad (3)$$

The first term is the initial test accuracy of round $t$; the second ($\Delta_L^{(t)}$) is the average gain by **L**ocal training; the third ($\Delta_G^{(t)}$) is the gain by **G**lobal aggregation. *A negative $\Delta_L^{(t)}$ indicates that local models after local training (at round $t$) have somewhat "forgotten" what the global model $\bar{\boldsymbol{\theta}}^{(t-1)}$ has learned (Kirkpatrick et al., 2017).* Namely, the local models drift away from the global model.

**Analysis on the training dynamics.** We use the same setup as in subsection 4.3 on CIFAR-10 and Tiny-ImageNet: $M = 10$, $\alpha = 0.3$. Figure 6 summarizes the values defined in Equation 3. For each combination (dataset + pre-training or not), we show the test accuracy using the global model $\bar{\boldsymbol{\theta}}^{(t)}$ ($\times$). We also show the averaged test accuracy using each local model $\tilde{\boldsymbol{\theta}}_m^{(t)}$ ($\bullet$). The red and green arrows indicate the gain by local training ($\Delta_L^{(t)}$) and global aggregation ($\Delta_G^{(t)}$), respectively. For brevity, we only draw the first **50 rounds** but the final accuracy is by **100 rounds**.

We have the following observations. First, pre-training seems to not alleviate local model drifting (Li et al., 2020b; Karimireddy et al., 2020b). Both FEDAVG with and without pre-training have notable negative $\Delta_L^{(t)}$, which can be seen from the *slanting* line segments: segments with negative slopes (i.e., $\bullet$ of round $t$ is lower than $\times$ of round $t - 1$) indicate drifting. Second, pre-training seems to have a larger global aggregation gain $\Delta_G^{(t)}$ (vertical segments from $\bullet$ to $\times$), especially in early rounds.

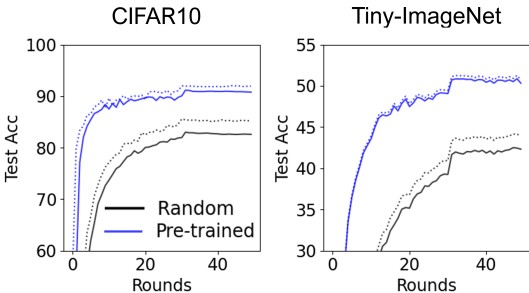

Figure 8: We show the test loss by FEDAVG's global model (curve) and the $95\%$ confidence interval of the test losses by the sampled convex combinations (shaded area). We also show the histograms of losses at $t = 10, 50$.

**Analysis on global aggregation.** We conduct a further analysis on global aggregation. In FEDAVG (Equation 2), global aggregation is by a simple weight average, using local data sizes as coefficients. According to (Karimireddy et al., 2020b), this simple average may gradually deviate from the solution of Equation 1 under non-IID conditions, and ultimately lead to a drastic accuracy drop. Here, we analyze if pre-training can alleviate this issue. As it is unlikely to find a unique minimum of Equation 1 to calculate the deviation, we propose an alternative way to quantify the robustness of aggregation.

Our idea is inspired by Chen & Chao (2021), who showed that the coefficients used by FEDAVG may not optimally combine the local models. This motivates us to search for the **optimal convex aggregation** and calculate its accuracy gap against the simple average. The smaller the gap is, the more robust the global aggregation is. We define the optimal convex aggregation as follows:

$$\bar{\theta}^{\star(t)} = \sum_{m=1}^{M} \lambda_m^\star \tilde{\theta}_m^{(t)}, \qquad \text{where } \{\lambda_m^\star\} = \arg\max_{\{\lambda_m \geq 0; \sum_m \lambda_m = 1\}} \mathcal{A}_{\text{test}}(\sum_{m=1}^{M} \lambda_m \tilde{\theta}_m^{(t)}). \qquad (4)$$

That is, we search for $\boldsymbol{\lambda} = [\lambda_1, \cdots, \lambda_M]^\top$ in the $(M-1)$-simplex that maximizes $\mathcal{A}_{\text{test}}$. We apply SGD to find $\{\lambda_m^\star\}$. (See the Suppl.)

Figure 7 shows the curves of $\mathcal{A}_{\text{test}}(\bar{\theta}^{(t)})$ and $\mathcal{A}_{\text{test}}(\bar{\theta}^{\star(t)})$. For $\mathcal{A}_{\text{test}}(\bar{\theta}^{\star(t)})$, we replace the simple weight average by the optimal convex aggregation throughout the entire FEDAVG: at the beginning of each round, we send the optimal convex aggregation back to clients for their local training. It can be seen that $\mathcal{A}_{\text{test}}(\bar{\theta}^{\star(t)})$ outperforms $\mathcal{A}_{\text{test}}(\bar{\theta}^{(t)})$. The gap is larger for FEDAVG without pre-training than with pre-training. *Namely, for FEDAVG with pre-training,*

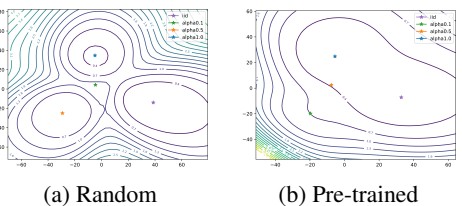

Figure 7: We show for each dataset (and pre-training or not) the test accuracy using the global model $\bar{\theta}^{(t)}$ (solid) and the optimal convex aggregation $\bar{\theta}^{\star(t)}$ (dashed).

*the simple weight average has a much closer accuracy to the optimal convex aggregation.*

**Analysis on the loss surface.** To see why pre-training leads to robust aggregation, we investigate the variation of $\mathcal{L}_{\text{test}}(\sum_{m=1}^{M} \lambda_m \tilde{\theta}_m^{(t)})$ across different $\boldsymbol{\lambda}$ on the simplex. We sample $\lambda$ for 300 times, construct the global models and calculate the test losses, and compute the $95\%$ confidence interval. As shown in Figure 8, FEDAVG with pre-training has a smaller interval; i.e., a lower-variance loss surface in aggregation. This helps explain why it has a smaller gap between $\mathcal{A}_{\text{test}}(\bar{\theta}^{(t)})$ and $\mathcal{A}_{\text{test}}(\bar{\theta}^{\star(t)})$.

We further visualize the loss landscapes (Li et al., 2018). We use the same random (or pre-trained) weights to initialize FEDAVG on CIFAR-10 ($M = 10$), for different $\alpha$ and for an IID condition. We then gather the final global models and project them onto the loss landscape of $\mathcal{L}$, i.e., the global empirical risk. We found that pre-training enables the global models under different data conditions (including the IID one) to converge to the same loss

(a) Random          (b) Pre-trained

Figure 9: **Visualizations of the loss landscape.**

basin (with a lower loss). In contrast, without pre-training, the global models converge to isolated, and often poor loss basins. This offers another explanation of why pre-training improves FL.

**Conclusion.** We conduct the very first systematic study on pre-training for federated learning (FL) to explore a rarely studied aspect: *initialization*. We show that pre-training largely bridges the gap between FL and centralized learning. We make an attempt to understand the underlying effects and reveal several new insights into FL, potentially opening up future research directions.

## ACKNOWLEDGMENTS

This research is supported in part by grants from the National Science Foundation (IIS-2107077, OAC-2118240, and OAC-2112606), the OSU GI Development funds, and Cisco Systems, Inc. We are thankful for the generous support of the computational resources by the Ohio Supercomputer Center and AWS Cloud Credits for Research. We thank all the feedback from the review committee and have incorporated them.

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

# SUPPLEMENTARY MATERIAL

We provide details omitted in the main paper.

- Appendix A: additional details of fractals pre-training (cf. section 4 of the main paper).
- Appendix B: details of experimental setups (cf. section 6 and section 4 of the main paper).
- Appendix C: additional experimental results and analysis (cf. section 6 of the main paper).
- Appendix D: additional discussions.

## A  ADDITIONAL DETAILS OF FRACTAL PRE-TRAINING

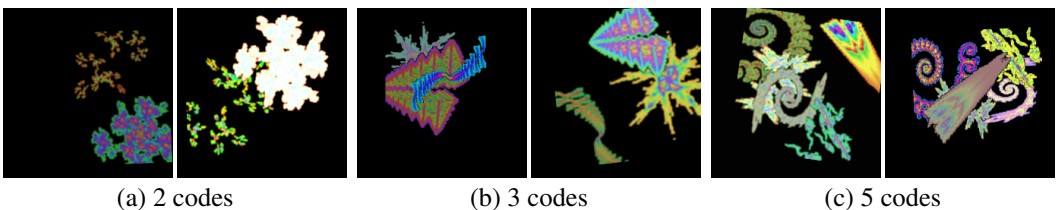

(a) 2 codes                (b) 3 codes                (c) 5 codes

Figure 10: **Examples of image pairs of FPS.** We generate three image pairs with FPS using $I = 2, 3$, and 5 IFS codes. The resulting pairs are shown in (a), (b), and (c), which contain 2, 3, and 5 fractals in each image, respectively. In each pair, the two fractals rendered from the same IFS code reflect intra-class variations and different placements.

### A.1  A COMPLETE INTRODUCTION OF OUR FRACTAL PAIR SIMILARITY (FPS)

In section 4 of the main paper, we only provide a condensed introduction of our FPS algorithm due to the page limits. Now, we give a complete version that elaborates more details.

#### A.1.1  BACKGROUND: SUPERVISED FRACTAL PRE-TRAINING

**Fractal generation.** A fractal image can be rendered via an affine Iterative Function System (IFS) Kataoka et al. (2020); Barnsley (2014). The IFS rendering process can be thought of as "drawing" points iteratively on a canvas. The point transition is governed by a set of $2 \times 2$ affine transformations, which we call an IFS code $\mathcal{S}$:

$$\mathcal{S} = \left\{ \left( \boldsymbol{A}_k \in \mathbb{R}^{2 \times 2}, \boldsymbol{b}_k \in \mathbb{R}^2, p_k \in \mathbb{R} \right) \right\}_{k=1}^{K}. \tag{5}$$

Here, an $(\boldsymbol{A}_k, \boldsymbol{b}_k)$ pair specifies an affine transformation and $p_k$ is the corresponding probability, i.e., $\sum_k p_k = 1$ and $p_k \geq 0, \forall k \in [K]$. Concretely, given $\mathcal{S}$, an IFS generates an image as follows. Starting from an initial point $\boldsymbol{v}_0 \in \mathbb{R}^2$, it repeatedly samples one transformation $k$ with replacement according to the probability $p_k$, and performs $\boldsymbol{v}_{n+1} = \boldsymbol{A}_k \boldsymbol{v}_n + \boldsymbol{b}_k$ to arrive at the next point. This stochastic process continues until a sufficient number of iterations is reached. The traveled points $\{\boldsymbol{v}_0, \cdots, \boldsymbol{v}_n, \cdots\}$ are then used to synthesize a fractal: by rendering each point as a binary or continuous value on a black canvas. Due to the randomness in the IFS, one code can create different but geometrically-similar fractals.

**Supervised pre-training.** While fractal images do not look real, they are diverse and capture the geometric properties of elements found in nature (Mandelbrot & Mandelbrot, 1982), thus suitable for pre-training and facilitating downstream tasks (Kataoka et al., 2020). According to Baradad et al. (2021), diversity and naturalism (i.e., capturing certain structural properties of real data) are two key properties that make for good synthetic data for training vision systems. Pre-training with fractal images thus allows the model to capture features, e.g., structured properties and the diversity of patterns, that can be useful for recognizing real data.

Since the IFS code controls the generation process and hence the geometry of the fractal, it can essentially be seen as the fractal's ID or class label. In (Kataoka et al., 2020), the authors proposed to sample $J$ different codes $\{\mathcal{S}_j\}_{j=1}^J$ and create for each code a number of images to construct a $J$-class

classification dataset. This synthetic labeled dataset is then used to pre-train a neural network via a multi-class loss (e.g., cross-entropy). The following-up work by Anderson & Farrell (2022) proposed to create *more complex images by painting multiple (denoted by $I$) fractals on one canvas*. The resulting image thus has $I$ out of $J$ labels, on which a multi-label loss is more suitable for supervised pre-training.

### A.1.2  OUR APPROACH: FRACTAL PAIR SIMILARITY (FPS)

We propose a novel way for fractal pre-training, inspired by one critical insight — *we can in theory sample infinitely many IFS codes and create fractals with highly diverse geometric properties*. That is, instead of creating a $J$-class dataset that limits the diversity of codes (e.g., $J =$1K) but focuses more on intra-class variation, we propose to trade the latter for the former.

We propose to sample a new set of codes and create an image that contains multiple fractals on the fly. Namely, for each image, we sample a small set of $I$ codes $\{\mathcal{S}_i\}_{i=1}^{I}$, generate $I$ corresponding fractals, and composite them into one image. The resulting dataset will have all its images from different classes (i.e., different sets of $\{\mathcal{S}_i\}_{i=1}^{I}$) in theory.

**Analogy to self-supervised learning.** *How can we pre-train a neural network using a dataset whose images are all of the different class labels?* Here, we propose to view the dataset as essentially "unlabeled" and draw an analogy to self-supervised learning Liu et al. (2021), especially those based on contrastive learning (Wu et al., 2018; He et al., 2020b; Chen et al., 2020d;b;c; Tian et al., 2020) or similarity learning Grill et al. (2020); Chen & He (2021). The core idea of these approaches is to treat every image as from a different class and learn to either repulse different images away (i.e., negative pairs) or draw an image and an augmented version of it closer (i.e., positive pairs). This line of approaches is so effective that they can even outperform supervised pre-training in many tasks Ericsson et al. (2021). *We note that while Baradad et al. (2021) has applied self-supervised learning to fractals, the motivation is very different from ours: the authors directly used the supervised dataset created by Kataoka et al. (2020) but ignored the labels.*

**Fractal Pair Similarity (FPS).** Conventionally, to employ contrastive learning or similarity learning, one must perform data augmentation such that a single image becomes a positive pair. Common methods are image-level scale/color jittering, flips, crops, RandAugment Cubuk et al. (2020), Self-Augment Reed et al. (2021), etc. Here, *we propose to exploit one unique way for fractals (or broadly, synthetic data), which is to use the IFS to create a pair of images based on the same set of codes* $\{\mathcal{S}_i\}_{i=1}^{I}$. We argue that this method can create stronger and more physically-meaningful argumentation: not only do different fractals from the same codes capture intra-class variation, but we also can create a pair of images with different placements of the same $I$ fractals (more like object-level augmentation). Moreover, this method can easily be compatible with commonly used augmentation.

**Implementation.** We detail the generation of a positive image pair in FPS as follows. First, we randomly sample $I$ distinct IFS codes. Then, each IFS code is used to produce two fractal shapes applied with random coloring, resizing, and flipping. Finally, we obtain two sets of fractals; each set contains $I$ distinct fractal shapes that can be pasted on a black canvas to generate one fractal image. The resulting two fractal images are further applied with image augmentations, such as random resized cropping, color jittering, flipping, etc. We provide more examples of image pairs generated by our FPS in Figure 10. In each pair, the two fractals generated from the same IFS code show intra-class variations, in terms of shapes and colors, and are placed at random locations.

We then apply self-supervised learning algorithms for pre-training. We use the IFS code sampling scheme proposed in Anderson & Farrell (2022) (with $K \in \{2, 3, 4\}$) and similarly apply scale jittering on each IFS code before we use it to render fractals. For efficiency, we pre-sample a total of 100K IFS codes in advance and uniformly sample $I$ codes from them to generate a pair of images.

Our PyTorch code implementation is provided at `https://github.com/andytu28/FPS_Pre-training`.

### A.2  SELF-SUPERVISED LEARNING APPROACHES

After generating fractal image pairs with FPS, we pre-train models by applying two self-supervised learning approaches, SimSiam Chen & He (2021) (similarity-based) and MoCo-V2 Chen et al. (2020d) (contrastive-based). We briefly review them in the following.

Table 8: Summary of datasets and setups.

| Dataset | Task | #Class | #Training | #Test/Valid | Split | #Clients | Reso. | Networks |
|---|---|---|---|---|---|---|---|---|
| CIFAR-10/100 | Classification | 10/100 | $50K$ | $10K$ | Dirichlet | 10 | $32^2$ | ResNet-$\{20, 32, 44, 56\}$ |
| Tiny-ImageNet | Classification | 200 | $100K$ | $10K$ | Dirichlet | 10/100 | $64^2$ | ResNet-18 |
| iNaturalist-2017 | Classification | 1203 | $120K$ | $36K$ | GEO-10K/3K | 39/136 | $224^2$ | ResNet-18 |
| Cityscapes | Segmentation | 19 | $3K$ | $0.5K$ | Cities | 18 | $768^2$ | DeepLabV3 + MobileNet-V2 |

**SimSiam.** Given a positive image pair $(\boldsymbol{x}_1, \boldsymbol{x}_2)$, we process them by an encoder network $f$ to extract their features. A prediction MLP head $h$ is then applied to transform the features of one image to match the features of the other. Let $\boldsymbol{p}_1 = h(f(\boldsymbol{x}_1))$ and $\boldsymbol{z}_2 = f(\boldsymbol{x}_2)$. The objective (to learn $f$ and $h$) is to minimize the negative cosine similarity between them:

$$D(\boldsymbol{p}_1, \boldsymbol{z}_2) = -\frac{\boldsymbol{p}_1}{||\boldsymbol{p}_1||_2} \cdot \frac{\boldsymbol{z}_2}{||\boldsymbol{z}_2||_2}, \tag{6}$$

where $|| \cdot ||_2$ is the $l_2$-norm. Since the relation between $\boldsymbol{x}_1$ and $\boldsymbol{x}_2$ is symmetric, the final loss can be written as follows:

$$\mathcal{L} = \frac{1}{2}D(\boldsymbol{p}_1, \boldsymbol{z}_2) + \frac{1}{2}D(\boldsymbol{p}_2, \boldsymbol{z}_1). \tag{7}$$

**MoCo-V2.** Similar to SimSiam Chen & He (2021), MoCo-V2 Chen et al. (2020d) also aims to maximize the similarity between features extracted from a positive image pair $(\boldsymbol{x}_1, \boldsymbol{x}_2)$. The main difference is that MoCo-V2 adopts contrastive loss, which also takes negative pairs into account. Following the naming in MoCo-V2 Chen et al. (2020d), let $\boldsymbol{x}^q = \boldsymbol{x}_1$ be the query and $\boldsymbol{x}^k_+ = \boldsymbol{x}_2$ be the positive key. We also have negative images/keys $\{\boldsymbol{x}^k_1, \boldsymbol{x}^k_2, ...\boldsymbol{x}^k_N\}$ in the mini-batch. We define $\boldsymbol{q} = f_q(\boldsymbol{x}^q)$, $\boldsymbol{k}_+ = f_k(\boldsymbol{x}^k_+)$, and $\boldsymbol{k}_i = f_k(\boldsymbol{x}^k_i)$, where $f_q$ is the encoder for query images and $f_k$ is the encoder for keys. The objective function for MoCo-V2 (to learn $f_q$ and $f_k$) is written as follows:

$$\mathcal{L} = -\log \frac{\exp(\boldsymbol{q} \cdot \boldsymbol{k}_+/\tau)}{\sum_{i=0}^{N} \exp(\boldsymbol{q} \cdot \boldsymbol{k}_i/\tau)}, \tag{8}$$

where $\tau$ is a temperature hyper-parameter. Besides the contrastive loss, MoCo-V2 maintains a dictionary to store and reuse features of keys from previous mini-batches, thereby making the negative keys not limited to the current mini-batch. Finally, to enforce stability during training, a momentum update is applied on the key encoder $f_k$.

In subsection 4.5 and Table 6 of the main paper, we pre-train ResNet-18 and ResNet-20 for 100 epochs using SimSiam Chen & He (2021) and MoCo-V2 Chen et al. (2020d). Specifically, these neural network architectures are used for $f$ in SimSiam and $f_q$ and $f_k$ in MoCo-V2. Each epoch consists of 1M image pairs, which are generated on-the-fly, for FPS. For a comparison to StyleGAN in Table 6, we use the pre-generated StyleGAN dataset provided in Baradad et al. (2021), which has 1.3M images. After pre-training, we keep $f_q$ (discard $f_k$) from MoCo-V2, following Chen et al. (2020d), and keep $f$ (discard $h$) from SimSiam, following Chen & He (2021).

For SimSiam, we use the SGD optimizer with learning rate 0.05, momentum 0.9, weight decay 1e−4, and batch size 256. For MoCo-V2, we use the SGD optimizer with learning rate 0.03, momentum 0.9, weight decay 1e−4, and batch size 256. The dictionary size is set to 65, 536. For both SimSiam and MoCo-V2, the learning rate decay follows the cosine schedule (Loshchilov et al., 2017).

For other experiments besides Table 6, we mainly use SimSiam for FPS. We use the same training setup as mentioned above (e.g., 100 epochs).

## B FL EXPERIMENT DETAILS

### B.1 DATASETS, FL SETTINGS, AND HYPERPARAMETERS

We train FEDAVG for **100 rounds**, with 5 local epochs and weight decay 1e−4. Learning rates are decayed by 0.1 every 30 rounds. Besides that, we summarize the training hyperparameters for each

Table 9: Default FL settings and training hyperparameters in the main paper.

| Dataset | Non-IID | Sampling | Optimizer | Learning rate | Batch size |
|---|---|---|---|---|---|
| CIFAR-10/100 | Dirichlet($\{0.1, 0.3\}$) | 100% | SGD + 0.9 momentum | 0.01 | 32 |
| Tiny-ImageNet | Dirichlet(0.3) | 100%/10% | SGD + 0.9 momentum | 0.01 | 32 |
| iNaturalist-2017 | GEO-10K, GEO-3K | 50%/20% | SGD + 0.9 momentum | 0.1 | 128 |
| Cityscapes | Cities | 100% | Adam | 0.001 | 16 |

of the federated experiments included in the main paper in Table 9. We always reserve $2\%$ data of the training set as the validation set for hyperparameter searching for finalizing the setups.

For pre-processing, we generally follow the standard practice which normalizes the images and applies some augmentations. CIFAR-10/100 images are padded 2 pixels on each side, randomly flipped horizontally, and then randomly cropped back to $32 \times 32$. For the other datasets with larger resolutions, we simply randomly cropped to the desired sizes and flipped horizontally following the official PyTorch ImageNet training script.

For the Cityscapes dataset, we use output stride $16$. In training, the images are randomly cropped to $768 \times 768$ and resized to $2048 \times 1024$ in testing.

*To further understand the effects of hyperparameters, we provide more analysis on Tiny-ImageNet in subsection C.4.*

### B.2 OPTIMAL CONVEX AGGREGATION

We provide more details for learning the optimal convex experiment in section 6. To find the optimal combinations for averaging clients' weights, we optimize $\lambda$ using the SGD optimizer, with a learning rate $1e - 5$ for 20 epochs (batch size 32) on the global test set. The vector $\lambda$ is $\ell 1$ normalized and each entry is constrained to be non-negative (which can be done with a softmax function in PyTorch) to ensure the combinations are convex. Since the optimization problem is not convex, we initialize $\lambda$ with several different vectors, including uniform initialization, and return the best (in terms of the test accuracy) as $\{\lambda_m^\star\}$ for Equation 4.

### B.3 FEDERATED SELF-PRE-TRAINING

In section 5, we consider self-pre-training by running a federated self-supervised learning algorithm on the same decentralized training data before starting supervised FEDAVG. We adopt the state-of-the-art algorithm Lubana et al. (2022), which is based on clustering. We follow their stateless configurations and run it with 5 local epochs each round, a 0.003 learning rate, and 32 batch size, for 50 rounds. We set $32/8$ local/global clusters for CIFAR-10 and $256/32$ for local/global clusters for Tiny-ImageNet.

We attribute the improvement by SP to several findings from different but related contexts. Gallardo et al. (2021); Yang & Xu (2020); Liu et al. (2021) showed that self-supervised learning is more robust in learning representations from class-imbalanced and distribution-discrepant data; Reed et al. (2022) showed that the SP procedure helps adapt the representations to downstream tasks. With that being said, federated self-supervised learning is itself an open problem and deserves more attention. (In our preliminary trials, the algorithm by Lubana et al. (2022) can hardly scale up to iNaturalist and Cityscapes. The validation accuracy by $k$-nearest neighbors hardly improves.)

## C ADDITIONAL EXPERIMENTS AND ANALYSES

### C.1 SCOPE AND MORE EXPERIMENTS IN NLP

In this paper, we focus on computer vision (CV). We respectfully think this should not degrade our contributions. First, CV is a big area. Second, while many federated learning (FL) works focus on CV, most of them merely studied classification using simple datasets like CIFAR. We go beyond them by experimenting with iNaturalist and Cityscapes (for segmentation), which are realistic and remain challenging even in centralized learning (CL). We believe our focused study in CV and

Table 10: Sent140 accuracy (100 rounds; local epoch= 5, 10% clients per round).

| Init. / Setting | FL | CL | $|\Delta_{\text{CL-FL}}|$ |
|---|---|---|---|
| Random | 75.4 | 80.5 | 5.1 |
| Pre-trained | 83.4 (+8.0) | 85.0 (+4.5) | 1.6 |

Table 11: Shakespear next-character prediction accuracy (30 rounds; local epoch= 10, 10% clients per round, 606 clients in total).

| Init. / Setting | FL | CL | $|\Delta_{\text{CL-FL}}|$ |
|---|---|---|---|
| Random | 46.4 | 59.1 | 12.7 |
| Pre-trained | 52.6 (+6.2) | 59.6 (+0.6) | 7.0 |

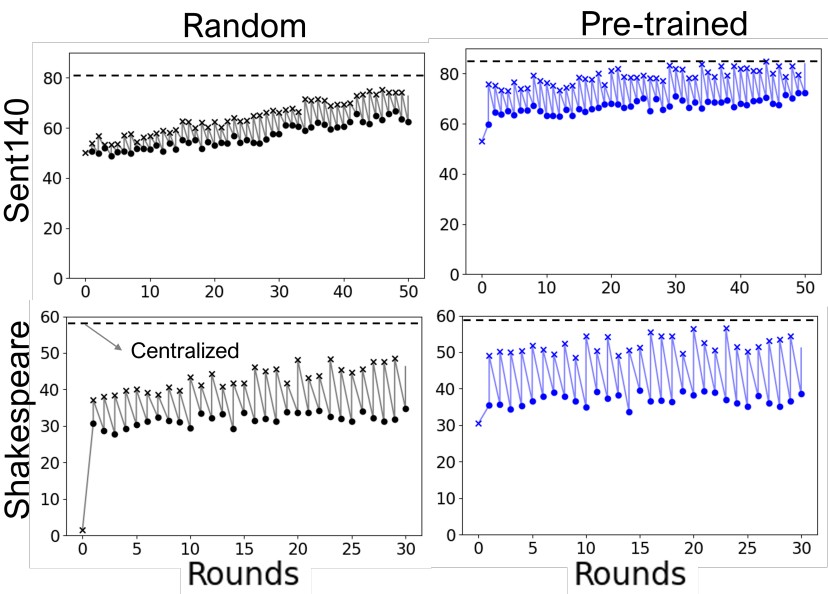

Figure 11: Dynamics of FL on Sent140 and Shakespeare. We show the test accuracy of the **global model** $\bar{\theta}^{(t)}$ ($\times$), and the averaged test accuracy using each **local model** $\tilde{\theta}_m^{(t)}$ ($\bullet$).

encouraging results on these two datasets using either real or synthetic pre-training are valuable to the FL community

That being said, here we provide two experiments on natural language processing (NLP) to verify that our observations are consistent with that on CV tasks. First, we conduct a sentiment analysis experiment on a large-scale federated Sent140 dataset (Caldas et al., 2019), which has 660K clients and 1.6M samples. We use a pre-trained DistilBERT (Sanh et al., 2019). Second, we experiment with another popular FL NLP dataset Shakespeare next-character prediction task proposed in McMahan et al. (2017). We use a version provided by (Caldas et al., 2019) which contains 606 clients, and use the Penn Treebank dataset (Marcinkiewicz et al., 1994) for pre-training an LSTM of two layers of 256 units each.

Table 10 and Table 11 show the results for the Sent140 and Shakespeare datasets, respectively. We also see a similar trend in Figure 11 — pre-training helps more in FL than in CL and largely closes their gap, even if the local model drifts are not alleviated.

## C.2 MORE ANALYSIS ON GLOBAL AGGREGATION

In section 6, we discuss how pre-training leads to more robust aggregation where the loss variance is smaller when we sample the convex combinations of local models. In Figure 8, we provide CIFAR-10 and Tiny-ImageNet due to the space limit. In Figure 12 and Figure 13, we also include the

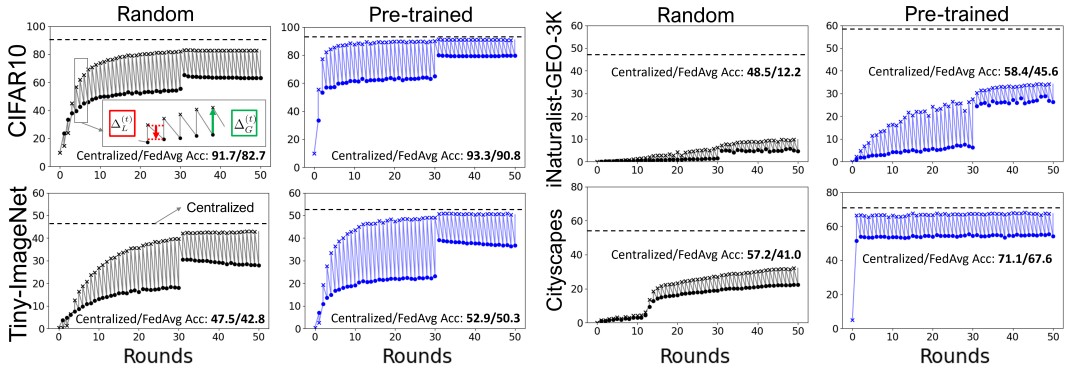

Figure 12: **Training dynamics of FEDAVG.** For each combination (dataset + pre-training or not), we show the test accuracy/mIoU using the **global model** $\bar{\theta}^{(t)}$ ($\times$), and the averaged test accuracy/mIoU using each **local model** $\tilde{\theta}_m^{(t)}$ ($\bullet$). The red and green arrows indicate the gain by local training ($\Delta_L^{(t)}$)/global aggregation ($\Delta_G^{(t)}$). For brevity, we only draw the first **50 rounds** but the final accuracy/mIoU are after **100 rounds**.

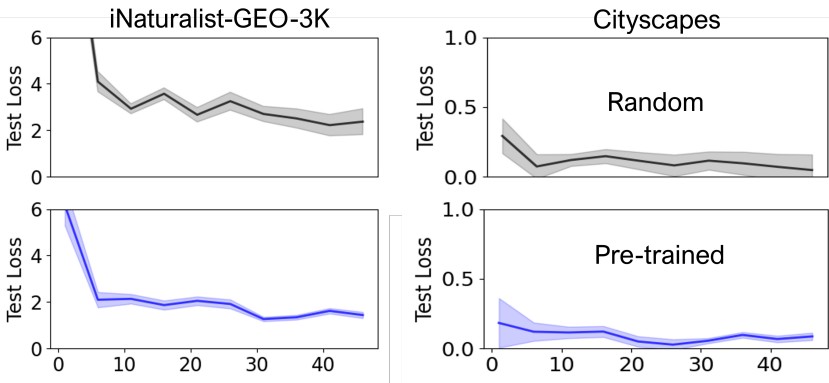

Figure 13: **Pre-training leads to a lower-variance loss surface** (cf. Figure 8). We show the test loss by the FEDAVG's global model $\bar{\theta}^{(t)}$ (curve) and the $95\%$ confidence interval of the test losses by the sampled convex combinations (shaded area) on iNaturalist-GEO-3K and Cityscapes.

iNaturalist-GEO-3K and Cityscapes. We have a consistent finding — using pre-training does not reduce local model drift but does lead to a lower-variance loss surface.

## C.3 MORE COMPARISONS ON SYNTHETIC PRE-TRAINING

In subsection 4.5, we compare synthetic pre-training methods. We also consider the images generated by (the best version of) a randomly-initialized StyleGAN Baradad et al. (2021); Karras et al. (2019). Due to the space limit, we only list some of the comparisons. Here we provide the complete results in Table 12. Across 3 datasets and 2 self-supervised algorithms, our FPS consistently outperforms the baselines. Since overall our FPS + SimSiam provides the strongest performance, we focus on it for the main paper. Here we also include the FPS + MoCo-V2 algorithm and observe a similar conclusion.

## C.4 DISCUSSIONS ON DIFFERENT FEDERATED SETTINGS

To understand how the FL settings affect the observations of pre-training, we further conduct studies on CIFAR-10 and Tiny-ImageNet in Figure 3 of the main text. We focus on the setting in the main text (i.e., 10 clients, Dir(0.3), 100% participation, local epoch= 5) but change one variable of settings such as learning rate scheduling (in Figure 14 and Figure 15), the number of clients, non-IID degree, participation rate, and the number of local epochs.

For different learning rate scheduling, we observe similar trends: pre-training does not alleviate client shift much but the test accuracy of the global model after aggregating the local models is higher.

Table 12: Comparison on synthetic pre-training methods in section 5. Means of 3 runs are reported.

| Init. / Dataset | C10 | C100 | Tiny-ImgNet |
|---|---|---|---|
| Random | 74.4±0.42 | 51.4±0.31 | 42.4±0.20 |
| Fractal + Multi-label | 73.0±0.43 | 51.0±0.44 | 40.9±0.22 |
| StyleGAN + SimSiam | 79.2±0.26 | 53.0±0.15 | 44.6±0.16 |
| Fractal + SimSiam | 77.4±0.30 | 51.7±0.28 | 44.2±0.20 |
| **FPS (ours)** + SimSiam | **80.5**±0.25 | **54.7**±0.11 | **45.7**±0.14 |
| StyleGAN + MoCo-V2 | 75.3±0.11 | 53.5±0.12 | 45.8±0.24 |
| Fractal + MoCo-V2 | 73.5±0.29 | 53.3±0.24 | 44.5±0.17 |
| **FPS (ours)** + MoCo-V2 | **79.9**±0.15 | **53.9**±0.16 | **46.1**±0.18 |
| ImageNet / Places365 | 87.9±0.11 | 62.2±0.09 | 50.3±0.08 |

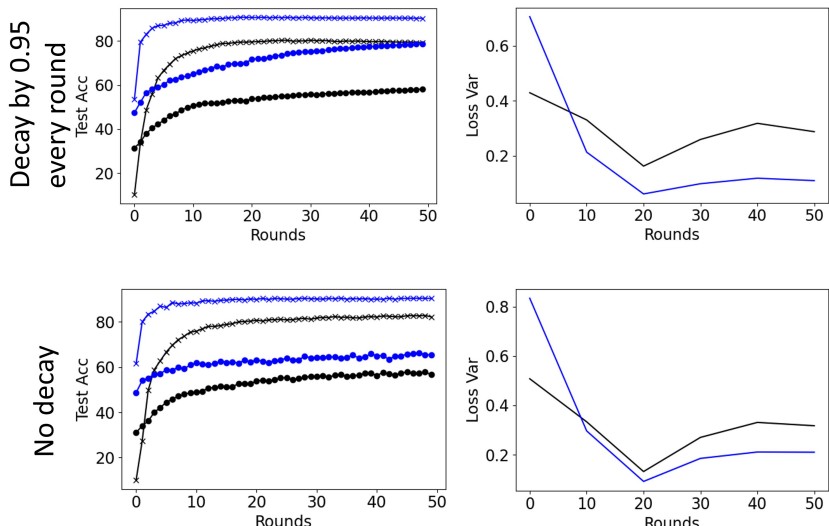

Figure 14: CIFAR-10 (10 clients, Dir(0.3), 100% participation, local epoch= 5) with different learning rate decay strategies. **Left**: we show **Global model** (×) accuracy and the averaged accuracy using each **local model** (●). **Right**: the test loss variance of 100 sampled coefficients (the width of the shaded area in Figure 8) to combine the local models in each round.

In the main paper, we focus on the schedule that decays the learning rate by $0.1$ every 30 rounds given its better performance compared to the two (decay by $0.95$ every round and no decays) here. In Figure 14 and Figure 15, we also monitor the variance of the test loss of the global models that use different coefficients (100 random samples) to aggregate the local models, i.e., the shaded area in Figure 8. We found the pre-trained ones have lower loss variance quite consistent across different learning rate scheduling and our analyses in section 6.

For client numbers, more clients will make the performance drop because each client has fewer data and the resulting local training is prone to over-fitting. However, using pre-training on both real data and our FPS still improves significantly.

For the non-IID degree, we manipulate it with the Dirichlet distributions by the $\alpha$ parameter. We observe that more non-IID (smaller $\alpha$) settings degrade the performance of using random initialization sharply while using pre-training is more robust (smaller accuracy drop).

For participation rates and the number of local epochs, we observe that they are not very sensitive variables, as long as they are large enough (e.g., participation rate $> 30\%$ and #local epochs $> 5$). Interestingly, using either fewer or more local epochs does not close the gap between using pre-training and random initialization.

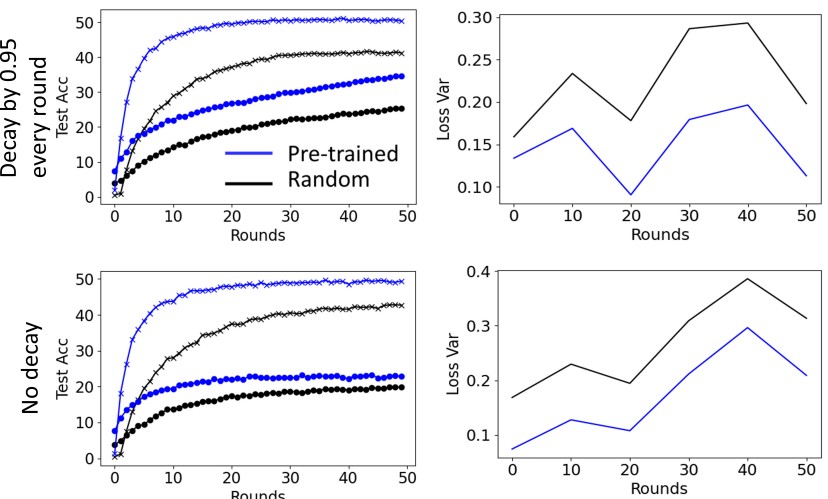

Figure 15: Tiny-ImageNet (10 clients, Dir(0.3), 100% participation, local epoch= 5) with different learning rate decay strategies. **Left**: we show **Global model** (×) accuracy and the averaged accuracy using each **local model** (●). **Right**: the test loss variance of 100 sampled coefficients (the width of the shaded area in Figure 8) to combine the local models in each round.

## C.5 ANALYSIS WITH FPS

For brevity, in section 6 of the main paper we mainly focus on using random initialization and using real data for pre-training. Here we also provide the analyses on global aggregation using our FPS, following section 6. We observe consistent effects on FL with FPS: it does not alleviate local model drift (Figure 16) but makes global aggregation more stable evidenced by smaller gains with optimal convex combinations (Figure 17) and lower loss variance (Figure 18).

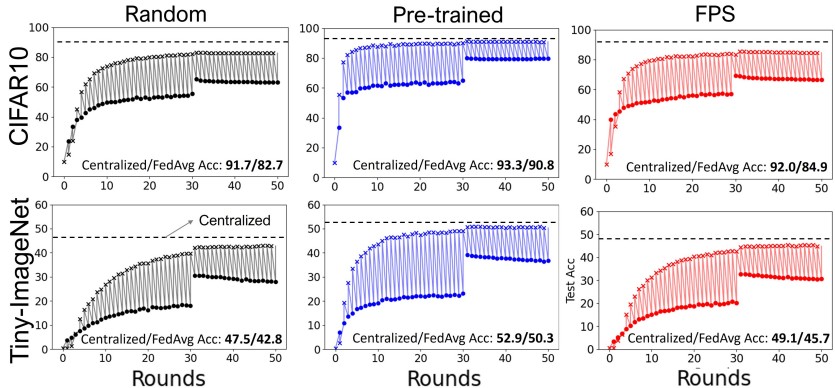

Figure 16: **Training dynamics of FEDAVG.** For each combination (dataset + pre-training or not), we show the test accuracy/mIoU using the **global model** $\bar{\theta}^{(t)}$ ($\times$), and the averaged test accuracy using each **local model** $\tilde{\theta}_m^{(t)}$ ($\bullet$). For brevity, we only draw the first **50 rounds** but the final accuracy/mIoU are after **100 rounds**. CIFAR-10 and Tiny-ImageNet (10 clients, Dir(0.3), 100% participation, local epoch= 5). The experiments follow section 6.

# D ADDITIONAL DISCUSSIONS

## D.1 POTENTIAL NEGATIVE SOCIETAL IMPACTS

Our work discusses how using pre-training on a large dataset can improve FL. While not specific to our discussions, collecting the real dataset could potentially inject data bias or undermine data privacy if the collection process is not carefully designed. However, we believe the concerns are mainly from data collection but not from the learning algorithms.

To remedy this, in section 5, we propose an alternative of using synthetic data that *does not require any real data* and can still improve FL significantly.

## D.2 COMPUTATION RESOURCES

We implement all the codes in PyTorch and train the models with GPUs. For experiments with $32 \times 32$ images, we trained on a 2080 Ti GPU. Pre-training takes about 1 day and FL takes about 8 hours. For experiments with $224 \times 224$ images, we trained on an A6000 GPU. Pre-training takes about $2 - 3$ days. and FL takes about 1 day. For the Cityscape dataset, we trained with an A6000 GPU for about 2 days.

## D.3 MORE DISCUSSION ON INITIALIZATION AND PRE-TRAINED MODELS IN FL

In subsection 4.1, we consider pre-training a model in the server and use it as the initialization for further federated training.

Another very different and complementary problem on model initialization compared to ours is about selecting part of the parameters of the global model to initialize each round of local training at each client. Several works Li et al. (2021a); Zhu & Sun (2021) proposed to find a sparse mask for each client to improve communication and computation efficiency, which is an orthogonal problem to ours.

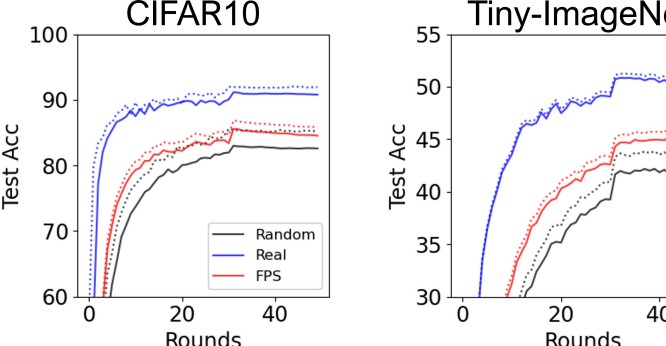

Figure 17: Optimal convex aggregation experiments on CIFAR-10 and Tiny-ImageNet (10 clients, Dir(0.3), 100% participation, local epoch= 5). The experiments follow Figure 7. We show for each combination (dataset + pre-training or not) the test accuracy using the global model $\bar{\boldsymbol{\theta}}^{(t)}$ (solid). We also show $\bar{\boldsymbol{\theta}}^{\star(t)}$ (dashed), which applies the optimal convex aggregation throughout the entire FEDAVG process. FEDAVG with pre-training has a smaller gap.

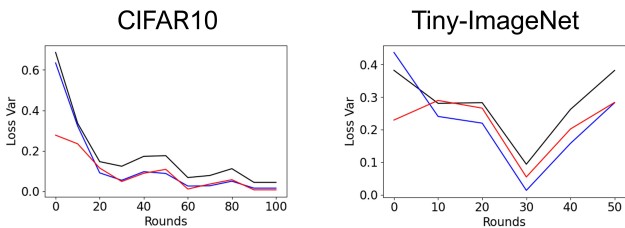

Figure 18: The loss variance of 100 sampled coefficients on CIFAR-10 and Tiny-ImageNet (10 clients, Dir(0.3), 100% participation, local epoch= 5). **Slightly different from Figure 8, here we plot the variance, which corresponds to the width of the shaded area in Figure 8**, for better visualization.

A recent work (Tan et al., 2022) instead uses several frozen pre-trained models in FL and only learns the class prototypes for classification.

## D.4 SYNTHETIC DATA FOR FL

Pre-training on synthetic data is a relatively new and fast-growing area in centralized learning. In our paper, we focus on generating synthetic data for images and using them for pre-training for initializing FL. We have shown the superiority of our proposed FPS against other synthetic data generation methods for pre-training, including one based on random StyleGANs Baradad et al. (2021); Karras et al. (2019). Besides vision tasks, we found many recent works showing the benefits of pre-training on carefully-designed synthetic data for downstream tasks on various modalities such as text (Wu et al., 2022), SQL tables (Jiang et al., 2022), speech (Fazel et al., 2021), etc. We believe it is promising to bring these methods into the federated setting and our study can be a valuable reference.

Another work FedSyn (Behera et al., 2022) considers training a generator (e.g., GAN) in a federated setting using clients' data, which is fairly different from our problem of generating synthetic data (without looking at clients' data) at the server for pre-training the global model.

