# OpenReview forum: "On the Importance and Applicability of Pre-Training for Federated Learning"
_ICLR.cc/2023/Conference — ICLR 2023 poster_

### Official Review · Reviewer_n7Y1 · 2022-10-17

**Confidence:** 3
**Correctness:** 3
**Technical Novelty And Significance:** 2
**Empirical Novelty And Significance:** 2
**Recommendation:** 6

**Clarity, Quality, Novelty And Reproducibility:**

- **Clarity**: The writing is clear. However, it is still unclear to me what is the main contribution of this work.
- **Quality**: The quality could be improved if the paper was organized differently (see the summary below.)
- **Novelty**: The presented work can have some novelty (FPS, a new self-supervised pretraining method for FL), which was however not very much emphasized.
- **Reproducibility**: Although implementation details are provided in the appendix, fully reproducing results is not very easy due to the lack of a code submission.

**Strength And Weaknesses:**

## Strong points

- The paper is well written and easy to read.
- The proposed work is well motivated.
- Experiments are systematic and extensive, demonstrating the effect of pretrained models from various perspectives.

## Weak points
Although the paper is largely well written and the motivation of the work is understandable, the main contribution is still unclear to me. I believe that this paper could have become stronger if it focused on how to effectively perform pretraining for federated learning on visual recognition tasks, rather than claiming to be "the very first systematic study on pre-training for FL".

- The use of pretraining for federated learning is not a new idea. At least in NLP, the use of pretrained models is well studied [a, b]. Especially [b] already showed that fine-tuning pretrained models with federated learning improved performances in both iid and non-iid settings. Given those existing works, the fact that pretraining worked well also for visual recognition tasks does not seem to be a very strong finding.
  - [a] Tian et al., "FedBERT: When Federated Learning Meets Pre-training", ACM Transactions on Intelligent Systems and Technology, 2022 https://dl.acm.org/doi/full/10.1145/3510033
  - [b] Weller et al., "Pretrained Models for Multilingual Federated Learning", NAACL, 2022 https://aclanthology.org/2022.naacl-main.101.pdf
- As the pretraining is already confirmed effective for FL in NLP, the next key question, at least to me, should be whether existing pretraining approaches are directly applicable or they should be extended in some way to be adapted to FL in CV. Indeed, the work proposed a new self-supervised pretraining algorithm called fractal pair similarity (FPS), very briefly in Section 4, as the authors found existing methods not effective. I believe that this paper could have become much more informative if this point was studied in more depth. Currently, Section 4.5 and Table 6 just showed that FPS combined with SimSiam performed the best on CIFAR10/100 and TinyImageNet. Further details and results are provided in Section A.1, C.3, and Table 12, which I believe should be more emphasized in the main body. Additionally, it also remains unclear if the same approach could be effective for various FL methods such as FedProx (Li et al., 2020b) and SCAFFOLD (Karimireddy et al., 2020a) presented in Section 2.

**Summary Of The Paper:**

This paper presents an empirical study to investigate the effectiveness of pretraining for federated learning. On the image recognition domain, models pre-trained with ImageNet, Place365, or synthetic images are tested. Synthetic images are used in pretraining with multi-label supervision or contrastive learning. An extensive experimental evaluation has revealed that the use of pretrained models brought significant performance boosts compared to when not doing so, on multiple datasets and data splits.


**Summary Of The Review:**

Overall, my major concern is the clarity of the paper regarding what is the main contribution of the proposed work. As the pretraining for FL has already been found effective in NLP, the current introduction seems to be a bit overclaiming in some points, such as "a different and rarely studied dimension in FL — model initialization", "We conduct the very first systematic study in these aspects, using visual recognition as the running example." Nevertheless, later the authors found that just how to enable pretraining for visual recognition on FL is not trivial, and motivated to develop a new method (FPS). I believe that that should be a potential contribution to be studied and emphasized more. If the proposed FPS is combined with multiple FL algorithms (e.g., FedAvg, FedProx, SCAFFOLD, etc.) and validated at the current level of experimentation and analysis for multiple tasks (e.g., classification, segmentation), I would be willing to increase the score.

---

> ### Author Response · Authors · 2022-12-07
> **Thank you again for the detailed feedback**
>
> Dear reviewer:
>
> We would like to thank you again for your efforts in reviewing our submission.
>
> We have tried our best to respond to most if not all of your questions. If our responses address your concerns, we sincerely hope you could reconsider the scores. We will also be very happy to answer any follow-up questions. We look forward to your updates.

---

> > ### Comment · Reviewer_n7Y1 · 2022-12-07
> > **thanks for the follow-up feedback**
> >
> > Thank you for the follow-up feedback, which has resolved a part of my initial concerns.
> >
> > That said, I still have a mixture feeling about the submission. My original comment was:
> >
> > > If the proposed FPS is combined with multiple FL algorithms (e.g., FedAvg, FedProx, SCAFFOLD, etc.) and validated at the current level of experimentation and analysis for multiple tasks (e.g., classification, segmentation), I would be willing to increase the score.
> >
> > On the one hand, the paper performed an extensive experiment comprising not only classification but segmentation, but only for FedAvg (correct me if I'm misunderstanding something). On the other hand, various FL algorithms are evaluated in Fig 6, but only for classification tasks.
> >
> > Personally, I believe that the main contribution of this paper would be to introduce FPS as an effective means of pre-training for FL, while doing an extensive and systematic study can be rather a side contribution. Nevertheless this is rather like a paper organization issue and could be mitigate by revising the introduction. Why is there no clear introduction of FPS in the Introduction section?
> >
> > I'll upgrade my score, but am not yet confident enough to strongly recommend acceptance.

---

### Official Review · Reviewer_acWZ · 2022-10-24

**Confidence:** 4
**Correctness:** 4
**Technical Novelty And Significance:** 3
**Empirical Novelty And Significance:** 4
**Recommendation:** 8

**Clarity, Quality, Novelty And Reproducibility:**

Clarity: The introduction of pre-train on FL is clear and provides sufficient experimental validation.
Quality: The first systematic study on pre-training for FL, and further reveal new insights into FL. The quality of the paper is high.
Novelty: This paper first systematic study on pre-training for FL. The paper has high novelty and originality.
Reproducibility: The author discusses in detail all the experimental methods in the paper, so the reproducibility is high.

**Strength And Weaknesses:**

Strength:
1. The first systematic study on pre-training for FL, using five image datasets including Cityscapes.
2. The authors further reveal new insights into FL, opening up future research directions.
According to abundant experiments, the authors find that pre-training largely bridges the gap between FL and centralized learning.
3. The experiment and analysis of the pre-training method on FL are sufficient.

Weakness:
1. When pre-training on synthetic data, the fractal image is quite different from the real sample image (such as ImageNet). Why is it also effective in testing datasets after pre-training?
2. The model adopted by the authors is mainly used for the classification task, and the network layer is not too deep, such as ResNet18 and ResNet20. If you pre-train on a deeper network model such as the ResNet50 or GoogleNet network, how does it work on FL?
3. The amount of data in the test set is small, such as 10K,0.5K,36K, etc. It is recommended to validate the advantage of pre-train in FL on a larger test set.

**Summary Of The Paper:**


Pre-training is prevalent in nowadays deep learning to improve the learned model’s performance. But, in federated learning (FL), neural networks are mostly initialized with random weights without pre-training weights.
The systematic study on pre-training for FL on five image datasets including Cityscapes.
According to abundant experiments, the authors find that pre-training largely bridges the gap between FL and centralized learning.


**Summary Of The Review:**

This  paper is good and can be accepted.

---

> ### Author Response · Authors · 2022-12-07
> **Thank you again for the detailed feedback**
>
> Dear reviewer:
>
> We would like to thank you again for your efforts in reviewing our submission.
>
> We have tried our best to respond to most if not all of your questions. We will be very happy to answer any follow-up questions. Thanks.

---

### Official Review · Reviewer_N8td · 2022-10-27

**Confidence:** 4
**Clarity, Quality, Novelty And Reproducibility:** The idea is interesting and details a…
**Correctness:** 3
**Technical Novelty And Significance:** 2
**Empirical Novelty And Significance:** 2
**Recommendation:** 5

**Strength And Weaknesses:**

Strengths:
The paper discusses the impact of pre-training in federated settings
The experiments show that the effectiveness of pre-training increases in challenging situations when there are higher non-IID degrees.
The authors also showed that pre-training with synthetic data could help in privacy-critical domains.

Weaknesses:
There is no literature on the impact of initialization in federated settings. Authors should compare the pre-training initialization with initialization techniques like FedMask.
Authors should compare pre-training performance using FedAVG with SGD and FedAvg with GD. This would help in understanding the impact of pre-training on communication.
The authors should compare the existing federated algorithms with the FedAdam algorithm.
There is no experiment on the impact of pre-training on system heterogeneity.
Authors should experiment with the impact of pre-training on convergence speed.
The authors showed an analysis of loss curves only for IID data. Is the initial loss for the model initialized with pre-trained weights less than the model initialized with random weights?
The major focus throughout the paper is on FedAvg, and analysis is reported on that only. No comparative analysis is reported for IID, and Non-IID cases with other existing FL approach.
Data at each client is generated differently to update the global model. Is there any impact of the pretraining on the convergence of each model concerning the data size (data heterogeneity)?
In the existing work for studying the effect of the pretraining on FL, authors have experimented with a higher value of M. However, authors in this paper have restricted to the 10.
The authors have proposed a method to generate synthetic data. However, in FL settings, multiple methods are introduced in existing works which are missing the paper, such as [1]. This follows no comparison of the pretraining with synthetic data generation methods.
[1] Behera, Monik Raj, et al. "FedSyn: Synthetic Data Generation using Federated Learning." arXiv preprint arXiv:2203.05931 (2022).
*Note: arxiv version is available at https://arxiv.org/pdf/2206.11488.pdf


**Summary Of The Paper:**

The paper discusses the applicability of pre-training in a federated learning setup. The authors did extensive experiments to demonstrate that pre-training can help close the gap in the performance difference between centralised and federated learning. The authors have shown that even pre-training with synthetic data improves performance in federated settings. The authors also show that pre-training helps the FedAvg algorithm tackle non-iid distribution. Moreover, the Authors also compare the FedAvg algorithms with different FL algorithms.


**Summary Of The Review:**

The reviewer is fairly confident that the evaluation is correct.

---

> ### Author Response · Authors · 2022-12-07
> **Thank you again for the detailed feedback**
>
> Dear reviewer:
>
> We would like to thank you again for your efforts in reviewing our submission.
>
> We have tried our best to respond to most if not all of your questions. If our responses address your concerns, we sincerely hope you could reconsider the scores. We will also be very happy to answer any follow-up questions. We look forward to your updates.

---

> ### Author Response · Authors · 2022-12-11
> **Thank you again for the detailed feedback**
>
> Dear reviewer:
>
> We would like to thank you again for your efforts in reviewing our submission.
>
> We have tried our best to respond to most if not all of your questions. If our responses address your concerns, we sincerely hope you could reconsider the scores. We will also be very happy to answer any follow-up questions. We look forward to your updates.

---

### Official Review · Reviewer_f2aE · 2022-10-29

**Confidence:** 3
**Clarity, Quality, Novelty And Reproducibility:** I think that it is no problem for tho…
**Correctness:** 4
**Technical Novelty And Significance:** 3
**Empirical Novelty And Significance:** 4
**Recommendation:** 8

**Strength And Weaknesses:**

[Strength]

S1. Federated learning is one of the significant topics in machine learning and artificial intelligence. Knowledge and insights for this topic will draw attention from a broad range of researchers and engineers.

S2. The problem dealt with in this paper is interesting. Applying pre-training to federated learning is thought to be difficult, mainly due to privacy concerns. This paper tackles this problem by introducing fractal-based image generators whose performances have already been proven in standard image recognition tasks.

S3. The significance of pre-training in federated learning has been examined and justified well by various kinds of empirical experimentation.

S4. The current manuscript is well-written and easy to follow.

[Weakness]

W1. I think that the current manuscript does not have any critical drawbacks. However, I am not sure whether we can provide synthetic data useful for federated learning in other (for example, audio, speech, and texts) domains.

**Summary Of The Paper:**

This paper deals with the problem of federated learning and demonstrates the significance of pre-training in this problem. More specifically, this paper chooses DEFAVG as the baseline for federated learning and considers using fractal-based image generators and self-supervised representation learning. The experimental results demonstrate that (1) pre-training improves the baseline, (2) performance gaps between federated and centralized learnings are getting smaller by pre-training, (3) those gaps become much smaller for more challenging settings, (4) pre-training is more effective for larger network models, and (5) pre-training by self-supervised learning with the same training examples as the primary network training can improve the performance of federated learning.

**Summary Of The Review:**

This paper can be accepted as-is.

---

### Public Comment · ~Yue_Tan2 · 2023-02-02
**Relation of This Paper to Prior Work**

Dear Authors,

Congratulations on your well written paper. I really appreciate the studies on FL by your team, i.e., FedBE, FedRoD, etc.

I would like to bring your attention to one of our prior works that also explores pre-training for FL. Your paper aims to understand how pre-training improves FL and how to effectively conduct pre-training for FL, while ours mainly focuses on integrating pre-trained models into FL and exploring how to effectively learn from pre-trained models in FL [1].

I think that "pre-training for FL" is quite an interesting and promising direction. Would you mind discussing this in your camera-ready version?

Thank you.

Yue Tan

[1] Federated Learning from Pre-Trained Models: A Contrastive Learning Approach. NeurIPS 2022.

---

> ### Author Response · Authors · 2023-02-03
> **Thank you**
>
> Hi Yue Tan,
>
> We appreciate your interest.
> We are glad to know about your exciting work and happy to cite it in our camera-ready paper.
>
> Best,
>
> Hong-You

---

### Decision · Program_Chairs · 2023-01-20

**Decision:**

Accept: poster

**Justification For Why Not Higher Score:**

The raised concerns by two reviewers about technical contributions and additional comparisons are legitimate (see above). While the paper passes the acceptance bar of ICLR, it should not be considered for a spotlight or oral presentation.

**Justification For Why Not Lower Score:**

N/A

**Metareview: Summary, Strengths And Weaknesses:**

The paper presents an interesting analysis of pre-training on federated learning considering real data, synthetic data, and client data. The analysis is carried out on multiple visual recognition datasets (classification/segmentation). Two reviewers recommend acceptance, one reviewer rates the paper as borderline reject and the other borderline accept. The concerns raised by the borderline reviewers are legitimate. As pointed out by Reviewer n7Y1, pre-training for federated learning has been studied and proven useful in NLP. Reviewer N8td suggested relevant additional comparisons, some of them addressed in the author response. The AC considers that these concerns, while sensible, are outweighed by the strengths of the paper as pointed out by the other two reviewers (and the author response), specifically the systematic analysis and insights provided with respect to pre-training on federated learning. Regarding the discussion about changing the story of the paper, the AC considers that the method for synthetic data pre-training is not the main contribution of the paper. Similar methods have been proposed before, for example see Baradad et al, Procedural Image Programs for Representation Learning (NeurIPS 2022). In the AC's view, the story should be kept the same, as the comprehensive study, analysis and insights are the main contributions of the paper.

**Note From Pc:**

if the above contains the word "oral" or "spotlight" please see: "oral" presentation means -> notable-top-5% and "spotlight" means -> notable-top-25%. As stated in our emails, we are disassociating presentation type from AC recommendations

**Summary Of Ac-Reviewer Meeting:**

N/A